# MATRIX SHUFFLE-EXCHANGE NETWORKS FOR HARD 2D TASKS

## ABSTRACT

Convolutional neural networks have become the main tools for processing two-dimensional data. They work well for images, yet convolutions have a limited receptive field that prevents its applications to more complex 2D tasks. We propose a new neural model, called Matrix Shuffle-Exchange network, that can efficiently exploit long-range dependencies in 2D data and has comparable speed to a convolutional neural network. It is derived from Neural Shuffle-Exchange network and has $\mathcal{O}(\log n)$ layers and $\mathcal{O}(n^2 \log n)$ total time and space complexity for processing a $n \times n$ data matrix. We show that the Matrix Shuffle-Exchange network is well-suited for algorithmic and logical reasoning tasks on matrices and dense graphs, exceeding convolutional and graph neural network baselines. Its distinct advantage is the capability of retaining full long-range dependency modelling when generalizing to larger instances – much larger than could be processed with models equipped with a dense attention mechanism.

## 1 INTRODUCTION

Data often comes in a form of two-dimensional matrices. Neural networks are often used for processing such data usually involving convolution as the primary processing method. But convolutions are local, capable of analyzing only neighbouring positions in the data matrix. That is good for images since the neighbouring pixels are closely related, but not sufficient for data having more distant relationships.

In this paper, we consider the problem how to efficiently process 2D data in a way that allows both local and long-range relationship modelling and propose a new neural architecture, called Matrix Shuffle-Exchange network, to this end. The complexity of the proposed architecture is $\mathcal{O}(n^2 \log n)$ for processing $n \times n$ data matrix, which is significantly lower than $\mathcal{O}(n^4)$ if one would use the attention (Bahdanau et al., 2014; Vaswani et al., 2017) in its pure form. The architecture is derived from the Neural Shuffle-Exchange networks (Freivalds et al., 2019; Draguns et al., 2020) by lifting their architecture from 1D to 2D.

We validate our model on tasks with differently structured 2D input/output data. It can handle complex data inter-dependencies present in algorithmic tasks on matrices such as transposition, rotation, arithmetic operations and matrix multiplication. Our model reaches the perfect accuracy on test instances of the same size it was trained on and generalizes on much larger instances. In contrast, a convolutional baseline can be trained only on small instances and does not generalize. The generalization capability is an important measure for algorithmic tasks to say that the model has learned an algorithm, not only fitted the training data.

Our model can be used for processing graphs by representing a graph with its adjacency matrix. It has a significant advantage over graph neural networks (GNN) in case of dense graphs having additional data associated with graph edges (for example, edge length) since GNNs typically attach data to vertices, not edges. We demonstrate that the proposed model can infer important local and non-local graph concepts by evaluating it on component labelling, triangle finding and transitivity tasks. It reaches the perfect accuracy on test instances of the same size it was trained on and generalizes on larger instances while the GNN baseline struggles to find these concepts even on small graphs. The model can perform complex logical reasoning required to solve Sudoku puzzles. It achieves 100% correct solutions on easy puzzles and 96.6% on hard puzzles which is on par with the state-of-the-art deep learning model which was specifically tailored for logical reasoning tasks.

## 2 RELATED WORK

Convolutional Neural Networks (CNN) are the primary tools for processing data with a 2D grid-like topology. For instance, VGG (Simonyan & Zisserman, 2014) and ResNet (He et al., 2016a) enable high-accuracy image classification. Convolution is an inherently local operation that limits CNN use for long-range dependency modelling. The problem can be mitigated by using dilated (atrous) convolutions that have an expanded receptive field. Such an approach works well on image segmentation (Yu & Koltun, 2015) but is not suitable for algorithmic tasks where generalization on larger inputs is crucial.

The attention mechanism (Bahdanau et al., 2014; Vaswani et al., 2017) is a widespread way of solving the long-range dependency problem in sequence tasks. Unfortunately, its application to 2D data is limited due to it's high $\mathcal{O}(n^4)$ time complexity for $n \times n$ input matrix. Various sparse attention mechanisms have been proposed to deal with the quadratic complexity of dense attention by attending only to a small predetermined subset of locations (Child et al., 2019; Beltagy et al., 2020; Zaheer et al., 2020). Reformer (Kitaev et al., 2020) uses locality-sensitive hashing to approximate attention in time $\mathcal{O}(n \log n)$. Linformer (Wang et al., 2020) uses a linear complexity approximation to the original attention by creating a low-rank factorization of the attention matrix. Sparse attention achieves great results on language modelling tasks, yet their application to complex data, where attending to entire input is required, is limited.

Graph Convolutional Neural Networks (Micheli, 2009; Atwood & Towsley, 2016) generalizes the convolution operation from the grid to graph data. They have emerged as powerful tools for processing graphs with complex relations (see Wu et al. (2019) for great reference). Such networks have successfully been applied to image segmentation (Gong et al., 2019), program reasoning (Allamanis et al., 2018), and combinatorial optimization (Li et al., 2018) tasks. Nonetheless, Xu et al. (2018) has shown that Graph Convolutional Networks may not distinguish some simple graph structures. To alleviate this problem they introduce Graph Isomorphism Network, that is as powerful as Weisfeiler-Lehman graph isomorphism test and is the most expressive among Graph Neural Network models.

Neural algorithm synthesis and induction is a widely explored topic for 1D sequence problems (Abolafia et al., 2020; Freivalds et al., 2019; Freivalds & Liepins, 2018; Kaiser & Sutskever, 2015; Draguns et al., 2020) but for 2D data, only a few works exist. Shin et al. (2018) has proposed Karel program synthesis from the input-output image pairs and execution trace. Differentiable Neural Computer (Graves et al., 2016), which employs external memory, has been applied to the SGRDLU puzzle game, shortest path finding and traversal tasks on small synthetic graphs. Several neural network architectures have been developed for learning to play board games (Silver et al., 2018), including chess (David et al., 2016) and go (Silver et al., 2016; 2017), often using complex architectures or reinforcement learning.

## 3 1D SHUFFLE-EXCHANGE NETWORKS

Here we review the Neural Shuffle-Exchange (NSE) network for sequence-to-sequence processing, recently introduced by Freivalds et al. (2019) and revised by Draguns et al. (2020). This architecture offers an efficient alternative to the attention mechanism and allows modelling of long-range dependencies in sequences of length $n$ in $\mathcal{O}(n \log n)$ time. NSE network is a neural adaption of well-known Shuffle-Exchange and Beneš interconnections networks, that allows linking any two devices using a logarithmic number of switching layers (see Dally & Towles (2004) for an excellent introduction).

The NSE network works for sequences of length $n = 2^k$, where $k \in \mathbb{Z}^+$, and it consists of alternating Switch and Shuffle layers. Although all the levels are of the same structure, a network formed of $2k - 2$ Switch and Shuffle layers can learn a broad class of functions, including arbitrary permutation of elements. Such a network is called a Beneš block. A deeper and more expressive network may be obtained by stacking several Beneš blocks; for most tasks, two blocks are enough.

The first $k - 1$ Switch and Shuffle layers of the Beneš block form Shuffle-Exchange block, the rest of the $k - 1$ Switch and Inverse Shuffle layers form its mirror counterpart. In the Beneš block, only Switch layers have learnable parameters and weight sharing is employed between layers of the same

Shuffle-Exchange block. A single Shuffle-Exchange block is sufficient to guarantee that any input is connected to any output through the network, i.e. the network has 'receptive field' spanning the whole sequence.

In the Switch layer, elements of the input sequence are divided into adjacent non-overlapping pairs, and the Switch Unit is applied to each pair. The Residual Switch Unit (RSU) proposed by Draguns et al. (2020) works the best. RSU has two inputs $[\boldsymbol{i}_1, \boldsymbol{i}_2]$, two outputs $[\boldsymbol{o}_1, \boldsymbol{o}_2]$, and two linear transformations on the feature dimension. After the first transformation, root mean square layer normalization (RMSNorm) (Zhang & Sennrich, 2019) and Gaussian Error Linear Unit (GELU) (Hendrycks & Gimpel, 2016) follow. By default, the hidden representation has two times more feature maps than the input. The second linear transformation is applied after GELU and its output is scaled by a scalar $h$. Additionally, the output of the unit is connected to its input using a residual connection that is scaled by a learnable parameter $\boldsymbol{s}$. RSU is defined as follows:

$$\boldsymbol{i} = [\boldsymbol{i}_1, \boldsymbol{i}_2]$$
$$\boldsymbol{g} = \text{GELU}(\text{RMSNorm}(\boldsymbol{Zi}))$$
$$\boldsymbol{c} = \boldsymbol{Wg} + \boldsymbol{b}$$
$$[\boldsymbol{o}_1, \boldsymbol{o}_2] = \sigma(\boldsymbol{s}) \odot \boldsymbol{i} + h \odot \boldsymbol{c}$$

In the above equations, $\boldsymbol{Z}$, $\boldsymbol{W}$ are weight matrices of size $2m \times 4m$ and $4m \times 2m$, respectively, where $m$ is the number of feature maps; $\boldsymbol{s}$ is a vector of size $2m$ and $\boldsymbol{b}$ is a bias vector – all of those are learnable parameters; $\odot$ denotes element-wise vector multiplication and $\sigma$ is the sigmoid function. $\boldsymbol{s}$ is initialized as $\sigma^{-1}(r)$ and $h$ is initialized as $\sqrt{1 - r^2} * 0.25$, where $r = 0.9$.

The Shuffle layer performs the perfect shuffle permutation of the sequence elements. In this permutation, the destination address of an element is obtained by cyclic bit rotation of its source address. The rotation to the right is used for the Shuffle layer and to the left for Inverse Shuffle layer. Shuffle layers do not have learnable parameters.

## 4 THE MATRIX SHUFFLE-EXCHANGE NETWORK

The naive way to employ NSE on 2D data would be by transforming an input matrix into a sequence using raster scan flattening (e.g., tf.reshape), but such a solution has two main drawbacks. Firstly, NSE requires more than 8 seconds for 1M element sequence (which matches flattened 1024x1024 matrix) in inference mode (Draguns et al., 2020), making its use unpractical on large matrices. Secondly, raster scan flattening doesn't preserve element locality if the inputs can be of different sizes, limiting the networks ability to generalize on large inputs – an essential requirement for algorithmic tasks.

We propose Matrix Shuffle-Exchange (Matrix-SE) network – an adoption of NSE for 2D data, that is significantly faster, generalizes on large matrices and retains the property of receptive field spanning the whole input matrix (see Appendix B for more details). The model is suitable for processing $n \times n$ input array, where $n = 2^k$ for $k \in \mathbb{Z}^+$ and each element is vector of $m$ feature maps. Inputs that don't fulfil this requirement has to be padded to the closest $2^k \times 2^k$ array.

The Matrix-SE network works by rearranging the input matrix into a 1D sequence, then applying several interleaved Quaternary Switch and Quaternary Shuffle layers and then converting data back to 2D, see Fig. 1. Rearranging of $n \times n$ input to $n^2$ sequence has to be done in such way that enables generalization on larger matrices. For this purpose, we read out values according to the Z-Order curve, as it has recursive structure and it preserves element locality regardless of the input size.[1] The resulting ordering is the same as one would get from a depth-first traversal of a quadtree. We call this transformation a Z-Order flatten. The left matrix in Fig. 2 shows matrix elements indexed according to Z-Order curve. The Z-Order unflatten transforms $n^2$ output sequence of the last Quaternary Switch layer to $n \times n$ representation according to Z-Order curve indexing.

The middle part involving Quaternary Switch (QSwitch) and Quaternary Shuffle (QShuffle) layers is structured as one or more Beneš blocks the same way as in the NSE network. The QSwitch and QShuffle layers differ from Switch and Shuffle layers in NSE, by performing the computations in groups of four instead of two. Such principle (using 4-to-4 switches instead of 2-to-2) has been proved for Shuffle-Exchange computer networks (Dally & Towles, 2004). That reduces network

---

[1]https://en.wikipedia.org/wiki/Z-order_curve

depth 2x times preserving the theoretical properties of NSE. Each Beneš block consists of interleaved $k-1$ QSwitch and QShuffle layers that form Shuffle-Exchange block, followed by $k-1$ QSwitch and Inverse QShuffle layers that form mirror counterpart of the first Shuffle-Exchange block. In contrast with NSE, we finalize Beneš block with one more QSwitch layer, making it more alike to Beneš computer networks. Last QSwitch layer serves as an output for the Beneš block and improves model expressiveness. Weight sharing is employed between QSwitch layers of same Shuffle-Exchange block. The last QSwitch layer of the Beneš block does not participate in weight sharing. If the network has more than one Beneš block, each block receives a distinct set of weights.

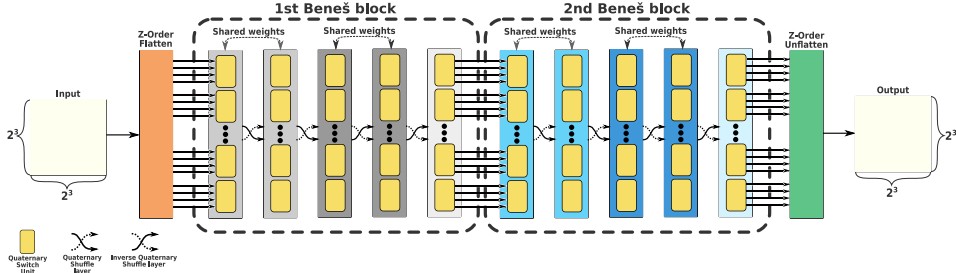

Figure 1: Matrix Shuffle-Exchange model consists of one or more Beneš block that is enclosed by Z-Order flatten, and Z-Order unflatten transformations. For a $2^k \times 2^k$ input array, Beneš block has total of $2k-1$ Quaternary Switch and $2k-2$ Quaternary Shuffle layers, giving rise to $\mathcal{O}(n^2 \log n)$ time complexity.

In the QSwitch layer, we divide adjacent elements of the sequence (corresponds to flattened 2D input) into non-overlapping 4 element tuples. That is implemented by reshaping the input sequence into a 4 times shorter sequence where 4 elements are concatenated along the feature dimension as $[\boldsymbol{i}^1, \boldsymbol{i}^2, \boldsymbol{i}^3, \boldsymbol{i}^4]$. Then Quaternary Switch Unit (QSU) is applied to each tuple. QSU is based on RSU but has 4 inputs $[\boldsymbol{i}^1, \boldsymbol{i}^2, \boldsymbol{i}^3, \boldsymbol{i}^4]$ and 4 outputs $[\boldsymbol{o}^1, \boldsymbol{o}^2, \boldsymbol{o}^3, \boldsymbol{o}^4]$. The rest of the Unit structure is left unchanged from RSU. The QSU is defined as follows:

$$\boldsymbol{i} = [\boldsymbol{i}_1, \boldsymbol{i}_2, \boldsymbol{i}_3, \boldsymbol{i}_4]$$
$$\boldsymbol{g} = \text{GELU}(\text{RMSNorm}(\boldsymbol{Z}\boldsymbol{i}))$$
$$\boldsymbol{c} = \boldsymbol{W}\boldsymbol{g} + \boldsymbol{b}$$
$$[\boldsymbol{o}_1, \boldsymbol{o}_2, \boldsymbol{o}_3, \boldsymbol{o}_4] = \sigma(\boldsymbol{s}) \odot \boldsymbol{i} + h \odot \boldsymbol{c}$$

In the above equations, $\boldsymbol{Z}, \boldsymbol{W}$ are weight matrices of size $4m \times 8m$ and $8m \times 4m$, respectively; $\boldsymbol{s}$ is a vector of size $4m$ and $\boldsymbol{b}$ is a bias vector – all of those are learnable parameters; $h$ is a scalar value; $\odot$ denotes element-wise vector multiplication and $\sigma$ is the sigmoid function. We initialize $\boldsymbol{s}$ and $h$ same as in RSU. Weights sharing is employed between all QSUs of the same QSwitch layer.

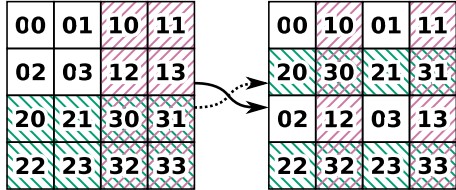

Figure 2: Depiction how the Quaternary Shuffle layer permutes the matrix (technically it permutes 1D sequence representation of matrix). The left image shows the matrix elements ordered according to Z-Order curve, where the ordering indices are represented by base-4 numbers. The right image shows element order after the Quaternary Shuffle. Quaternary Shuffle permutation can be interpreted as splitting matrix rows into two halves (white and green) and interleaving the halves, then applying the same transformation to columns (white and red).

The QShuffle layer rearranges the elements according to the cyclic digit rotation permutation. This permutation for a sequence $S$ is defined as $S[x] = S[qrotate(x, k)]$, where $qrotate(x, k)$ applies

cyclic shift by one digit to $x$ which is treated as a quaternary (base 4) number and $k$ is its length in quaternary digits.[2] For the first $k - 1$ QShuffle layers of the Beneš block, we apply rotation to the right – to the left for the remaining $k - 1$ layers.

## 5    EVALUATION

We have implemented our proposed architecture in TensorFlow. The code will be published on GitHub. All models are trained and evaluated on single Nvidia RTX 2080 Ti (11 GB) GPU using Rectified Adam (RAdam) optimizer (Liu et al., 2019) and softmax cross-entropy loss.

We evaluate the Matrix-SE model on several 2D tasks. Our main emphasis is on hard, previously unsolved tasks: algorithmic tasks on matrices and graphs and logical reasoning task – solving Sudoku puzzles. The proposed architecture is compared to ResNet and GNN baselines which are reasonable alternatives that could be used for these tasks. For algorithmic tasks, to achieve generalization, we use curriculum learning introduced by Freivalds & Liepins (2018) where models of different sizes are instantiated and trained simultaneously. We additionally evaluate the speed of the reported models and show that the Matrix-SE network achieves comparable or better results. Model is also evaluated on CIFAR-10 dataset and the results are given in Appendix E.

### 5.1    INFERRING MATRIX ALGORITHMS

We selected four matrix tasks of varying difficulty – transpose, rotation by 90 degrees, element-wise XOR of two matrices, and matrix squaring. For the transpose and rotation tasks, we generate a random input matrix from alphabet 1-11 and expect the network to learn the required element permutation. For the XOR task, we generate two binary matrices, place them side by side, add separator symbols between them, then apply padding to obtain a square matrix. All tasks, except matrix squaring, have simple $\mathcal{O}(n^2)$ algorithms.

Matrix squaring is the hardest of the selected tasks. We generate a random binary square matrix and ask the model to output its square (matrix multiplication with itself) modulo 2. That is a challenging task for the Matrix-SE model having the computation power of $\mathcal{O}(n^2 \log n)$, while the currently lowest asymptotic complexity for the matrix multiplication algorithm is $\mathcal{O}(n^{2.373})$ (Le Gall, 2014b). Nonetheless, fast approximation algorithm for matrix multiplication of $\tilde{\mathcal{O}}(n^2)$ complexity exists (Manne & Pal, 2014). The lower bound for matrix multiplication for real and complex numbers in the model of arithmetic circuits is $\Omega(n^2 \log n)$ proven by Raz (2002). Matrix squaring has the same complexity as matrix multiplication since these problems can be easily reduced one to the other. We selected matrix squaring for evaluation because it has only one input.

We evaluate two models – Matrix-SE network with 2 Beneš blocks, 96 feature maps having 3.5M learnable parameters and ResNet-29 model with 128 feature maps, kernel size of 3x3 having 4.5M learnable parameters (the model structure is given in Appendix C). Such ResNet-29 has a receptive field of size $59 \times 59$ that is enough to span the whole input on the largest matrix seen in training, making it on par with the Matrix-SE network that has a full receptive field for any input size.[3] Both models are trained on inputs up to size $32 \times 32$ for 500k steps and evaluated on size up to $1024 \times 1024$. Matrix-SE reaches perfect accuracy on the test examples of size up to $32 \times 32$ for all tasks, see Table 1. It generalizes on larger inputs for all tasks except matrix squaring. ResNet-29 struggles to learn these tasks even for size $32 \times 32$ and doesn't generalize. Matrix-SE also converges significantly faster than ResNet-29 (see Appendix D for more details).

### 5.2    INFERRING GRAPH ALGORITHMS

A graph can be easily represented by its adjacency matrix making Matrix-SE model a good fit for tasks on graphs, especially for tasks where edges have additional information. We consider three graph algorithmic tasks: connected component labelling, transitivity, and triangle finding.

For the connected component labelling task, we initialize a labelled graph with random edge labels in range 2-100. The task is to label all edges of each connected component with the lowest label among

---

[2]note that it matches the definition of $k$ above derived from $n$

[3]We used library proposed by Araujo et al. (2019) for calculating receptive field

Table 1: Element-wise test accuracy (average of 5 runs) on algorithmic tasks on matrices depending on input length. Both models are trained using curriculum learning on matrices up to size $32 \times 32$.

| Model | Task | Size of input matrix ($n \times n$) | | | | | | | | |
| | | Train size | | | | Generalization size | | | | |
| | | 4 | 8 | 16 | 32 | 64 | 128 | 256 | 512 | 1024 |
|---|---|---|---|---|---|---|---|---|---|---|
| Matrix-SE | Transpose | 1.0 | 1.0 | 1.0 | 1.0 | 1.0 | 1.0 | 1.0 | 1.0 | 1.0 |
| | Rotate by 90 | 1.0 | 1.0 | 1.0 | 1.0 | 1.0 | 1.0 | 0.8 | 0.41 | 0.19 |
| | Bitwise XOR | 1.0 | 1.0 | 1.0 | 1.0 | 0.96 | 0.89 | 0.78 | 0.67 | 0.56 |
| | Matrix squaring | 1.0 | 1.0 | 1.0 | 1.0 | 0.49 | 0.49 | 0.48 | 0.45 | 0.41 |
| ResNet-29 | Transpose | 1.0 | 1.0 | 1.0 | 0.99 | 0.36 | 0.15 | 0.08 | 0.06 | 0.05 |
| | Rotate by 90 | 1.0 | 1.0 | 1.0 | 0.99 | 0.17 | 0.08 | 0.06 | 0.06 | 0.05 |
| | Bitwise XOR | 1.0 | 1.0 | 1.0 | 1.0 | 0.4 | 0.31 | 0.28 | 0.26 | 0.26 |
| | Matrix squaring | 1.0 | 1.0 | 1.0 | 0.88 | 0.59 | 0.55 | 0.52 | 0.51 | 0.50 |

all the component's edges. For the transitivity task, the goal is to add edges to a directed graph for every two vertices that have a transitive path of length 2 between them. The hardest of the considered graph algorithmic tasks is the triangle finding. The fastest currently known algorithm relies on matrix multiplication and has asymptotic complexity of $\mathcal{O}(n^{2.373})$, where $n$ is the vertex count (Alon et al., 1997). Triangle finding is a fruitful field for quantum computing where an algorithm of quantum query complexity $\tilde{O}(n^{5/4})$ (Le Gall, 2014a) has been achieved. Therefore it is interesting if Matrix-SE can infer an $\mathcal{O}(n^2 \log n)$ time algorithm, at least approximately. For this task, we generate random complete bipartite graphs and add a few random edges. Although dense, such graphs have only a few triangles. The goal is to return all the edges belonging to any triangle.

Table 2: Element-wise test accuracy (average of 5 runs) on algorithmic tasks on graphs depending on vertex count. Both models are trained using curriculum learning on graphs with up to 32 vertices.

| Model | Task | Vertex count | | | | | | | |
| | | Train size | | | Generalization size | | | | |
| | | 8 | 16 | 32 | 64 | 128 | 256 | 512 | 1024 |
|---|---|---|---|---|---|---|---|---|---|
| Matrix-SE | Component Labeling | 1.0 | 1.0 | 1.0 | 1.0 | 0.99 | 0.97 | 0.96 | 0.91 |
| | Triangle Finding | 1.0 | 1.0 | 1.0 | 1.0 | 1.0 | 0.98 | 0.93 | 0.87 |
| | Transitivity | 1.0 | 1.0 | 1.0 | 0.96 | 0.88 | 0.82 | 0.77 | 0.71 |
| GIN | Component Labeling | 0.87 | 0.92 | 0.94 | 0.91 | 0.82 | 0.74 | 0.67 | 0.63 |
| | Triangle Finding | 0.91 | 0.95 | 0.95 | 0.87 | 0.51 | 0.63 | 0.77 | 0.89 |
| | Transitivity | 0.82 | 0.93 | 0.92 | 0.80 | 0.56 | 0.56 | 0.79 | - |

We use Matrix-SE network with 2 Beneš blocks, 192 feature maps and 14M learnable parameters (similar results can be achieved with 96 feature maps and 3.5M parameters). Since Graph Neural Networks are a natural choice for graph task, we also compare Matrix-SE with Graph Isomorphism Network (GIN) (Xu et al., 2018), that is proved (Xu et al., 2018) to be the most expressive among the message passing based Graph Neural Networks. For this purpose, we use GIN implementation from DGL library[4] with 5 Graph Isomorphism layers where a 3-layer perceptron is used at each layer. The final graph representation is mapped to the output using another 3-layer perceptron. The GIN network used in our experiments has 4.7M learnable parameters. Graph Neural Networks are more suitable for vertex-centric tasks, where all the features are represented in the vertices. To represent edge labels into GIN we follow a common practice (Yang et al., 2020; Veličković et al., 2017) and concatenate vertex features with element-wise multiplication between the edge and vertex features.

Both models are trained on graphs up to 32 vertices up to the moment where the loss hasn't decreased for 20k steps. Then models are tested on graphs with up to 1024 vertices. For the proposed tasks, Matrix-SE reaches perfect accuracy on test examples with up to 32 vertices and generalizes to moderately larger graphs. We find that Graph Isomorphism Network struggles to solve these tasks

---

[4]https://github.com/dmlc/dgl/tree/master/examples/pytorch/gin

even on graph sizes seen in training time. Also, GIN consumes more GPU memory than Matrix-SE – the graph for the transitivity task with 1024 vertices don't fit in GPU memory.

## 5.3 SOLVING SUDOKU

Solving a Sudoku puzzle (see Appendix A for more details) requires sophisticated reasoning and can be a challenging task for a human player. Since contemporary solvers, typically, use a backtracking search to solve Sudoku (Norvig, 2009; Crook, 2009), it's interesting to see if neural models are capable of such task without backtracking.

The complexity of a Sudoku puzzle is characterized by how many numbers are already given in the puzzle. Park (2018) has proposed solving easy Sudoku puzzles (24-36 givens) using a convolutional neural network. Wang et al. (2019) achieves close-to-perfect accuracy on slightly easier Sudoku puzzles (27-37 givens) by encoding each puzzle as MAXSat instance and then applying SATNet to it. Recurrent Relational Networks (Palm et al., 2018) can solve significantly harder Sudokus (17-34 givens) - they encode Sudoku puzzle as a graph and recursively employ Graph Neural Network for 32 steps. At each step, hidden representation is mapped to the solution by a 3-layer perceptron. Loss is obtained as the sum of a softmax cross-entropy loss of output at each step.

Table 3: Comparison of different neural Sudoku solvers. Sudoku puzzles with fewer givens are harder. The accuracy represents the fraction of completely correctly solved test puzzles.

| Model | Givens | Accuracy % |
|---|---|---|
| Matrix Shuffle-Exchange (this work) | 17-34 | 96.6 |
| Recurrent Relational Networks (Palm et al., 2018) | 17-34 | 96.6 |
| Matrix Shuffle-Exchange (this work) | 24-36 | 100.0 |
| Convolutional Neural Network (Park, 2018) | 24-36 | 70 |
| SATNet (Wang et al., 2019) | 27-37 | 98.3 |

We validate the Matrix-SE model on hard Sudoku dataset (17-34 givens) created by Palm et al. (2018) (see example from test set in Appendix A) and easy Sudoku dataset (24-36 givens) by Park (2018). Matrix-SE model is trained the same way as Recurrent Relational Network – we initiate a model with 2 Beneš blocks and 96 feature maps (3.5M total learnable parameters) and then apply it for 10 steps. At each step, hidden representation is mapped to output by one GELU layer and one linear layer then the softmax cross-entropy loss is applied. We represent a Sudoku puzzle in its natural form – as a matrix – and pad it to size $16 \times 16$. The model achieves 96.6% total accuracy on hard Sudoku dataset and 100% total accuracy on easy Sudoku dataset using 30 relational steps at evaluation, see Table 3. It's worth mentioning that Matrix-SE model requires fewer steps than Recurrent Relational Network (64 steps) and Sudoku doesn't need to be converted to a graph, significantly simplifying the training pipeline.

## 5.4 PERFORMANCE OF THE MODEL

Here we compare the training and evaluation speed of Matrix-SE network with other neural models. As a benchmark, we chose two algorithmic tasks – matrix squaring and graph transitivity task. The comparisons are done on single Nvidia RTX 2080Ti (11 Gb) GPU using Tensorflow. We report an average of 300 steps on a single element batch.

On the matrix squaring task, Matrix-SE (3.5M parameters) is compared to ResNet-29 (4.3M parameters) and Residual Shuffle-Exchange (RSE) network (3.5M parameters), see Fig. 3. Matrix-SE model works approximately 9x faster than RSE on large matrices and is on par with a convolutional neural network. It's worth mentioning that Matrix-SE is fast despite it has varying depth depending on the input size and its receptive field grows to cover the whole sequence while ResNet-29 has a fixed depth and receptive field.

To compare with graph networks, we chose the transitivity task for evaluation. We use Matrix-SE and Graph Isomorphism Network (GIN) of configurations given in Section 5.2. As depicted in Figure 4, the Matrix-SE model is slightly faster, both during training and evaluation, and it can process 2x larger dense graphs than GIN.

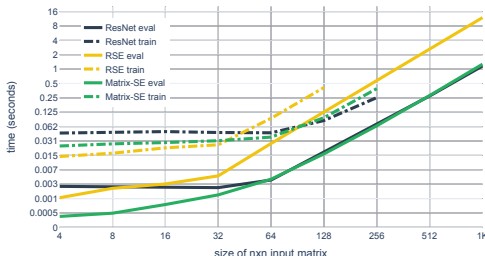 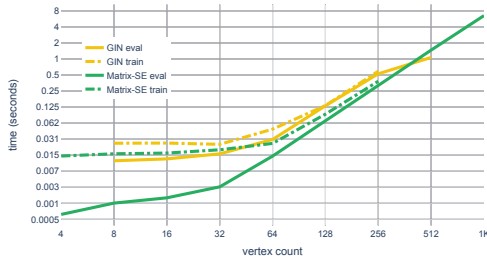

Figure 3: Training and evaluation speed on matrix squaring task for a single input instance.

Figure 4: Training and evaluation speed on graph transitivity task for a single input instance.

## 6 ABLATION STUDY

The most prominent feature that distinguishes the Matrix-SE from its predecessor – RSE is that it uses Z-Order flatten to transform data from a 2D matrix to 1D sequence. We study whether it is necessary by substituting Z-Order flatten with raster scan flatten. We train both versions on matrix squaring, transpose and triangle finding tasks for 500k steps (for triangle finding, first 150k are shown) on matrices of size up to $32 \times 32$. We can see in Fig. 6 that the version with Z-Order flatten generalizes to larger instances, and the other does not.

We also assess the effect of the model size in terms of feature map count and the model depth (Beneš blocks) on the convergence speed and accuracy. For the matrix squaring task, we use Matrix-SE with 2 Beneš blocks and 96 feature maps as a baseline, but, for triangle finding, Matrix-SE with 2 Beneš blocks and 192 feature maps. The results are depicted on Fig. 5, Fig. 7. The proposed baseline models work best for the chosen tasks. A larger and deeper model in most cases yield better results, but when reasonable accuracy is obtained, diminishing improvements are observed. On the matrix squaring and triangle finding tasks Z-Order flatten increase the convergence speed and accuracy compared to raster scan flatten.

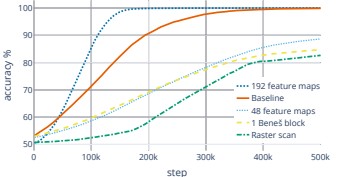 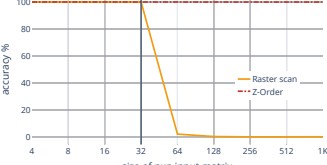 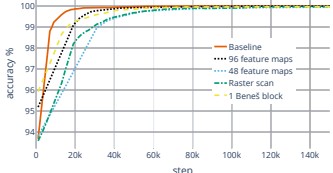

Figure 5: The effect of model size on the accuracy on the matrix squaring task.

Figure 6: Generalization impact of Z-Order flatten compared to raster scan flatten on matrix transitivity task.

Figure 7: The effect of model size on the accuracy for the triangle finding task.

## 7 DISCUSSION

We have introduced the Matrix Shuffle-Exchange model, which is of complexity $\mathcal{O}(n^2 \log n)$ and can model long-range dependencies in 2D data. It is capable of inferring matrix and graph algorithms purely from input-output examples exceeding convolutional and graph neural network baselines in terms of accuracy, speed and generalization power. It has shown considerable logical reasoning capabilities validated by ability to solve Sudoku puzzles, where it is on par with state-of-the-art neural networks.

This is the initial introduction of the Matrix Shuffle-Exchange model, and we further expect its significant improvements by fine-tuning its constituents, combination with other models and application to other tasks.

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

# A  SUDOKU PUZZLE

Sudoku is a popular logic-based puzzle is given as $9 \times 9$ board with at least 17 givens. The player is required to fill each row, column, and nine $3 \times 3$ subregions of the $9 \times 9$ board with a digit from 1 to 9. The same digit may not appear twice in the same row, column or subregion. McGuire et al. (2014) proved that there doesn't exist Sudoku puzzles with less than 17 givens, making them the hardest among all the classical Sudoku puzzles – easier puzzles can be obtained by providing more givens. Fig. 8 depicts an example of an unfilled Sudoku puzzle with 17 givens and its solution obtained by the Matrix Shuffle-Exchange network. For training, we used Sudoku dataset with hard puzzles having 17-36 givens (Palm et al., 2018). Training data was augmented by applying validity preserving transformations – transposition, stack[5] and band[6] permutations.

Figure 8: Sudoku puzzle with 17 given clues from the test set (Palm et al., 2018) and its solution obtained by the Matrix Shuffle-Exchange network.

---

[5]3 rows and 3 subregions

[6]3 columns and 3 subregions

# B    RECEPTIVE FIELD

The receptive field of a neural network is defined as the size of the region in the input that influences a single output feature. It has been proved that Beneš interconnection computer network with 4-to-4 switches and $2 \log n - 1$ switch and $2 \log n - 2$ shuffle layers for an input sequence of length $n^2$ can implement any input-output permutation (Dally & Towles, 2004). Since the Matrix Shuffle-Exchange network has the same interconnection structure as the Beneš computer network, it has the same theoretical properties. One can check that Matrix Shuffle-Exchange has the receptive field spanning the whole input by building a quaternary tree backwards from a single output element, as depicted in Fig. 9. Even after the first Shuffle-Exchange block, each input element participates in the construction of the feature.



Figure 9: The receptive field for the single output feature on $4 \times 4$ input and one Beneš block. Orange connections and green Switch Units depicts components that participate in the considered output feature; grey connections and Switch Units – ones that don't.

## C  RESNET-29 MODEL

It is known (Silver et al., 2016; 2017; 2018) that convolutional neural networks, more precise ResNet can estimate moves in many grid-based board games (e.g., chess, go, shogi). Because of that, ResNet seems a suitable choice for the algorithmic task on matrices. For all algorithmic tasks on matrices, we use pre-activation ResNet-29 (He et al., 2016b) with 3x3 convolutional kernels and 128 feature maps. Instead of Batch Normalization (Ioffe & Szegedy, 2015) we use Layer Normalization (Ba et al., 2016), as Batch Normalization gives bad results even on small matrices. The network consists of 14 residual blocks, that are prepended with embedding layer and 3x3 convolution. Fig. 10 depicts a residual block of the ResNet-29 network. The last residual block is followed by Layer Normalization, ReLU and a linear transformation. The network doesn't contain any bottlenecks or pooling layers. Fig. 11 depicts the full ResNet-29 structure.

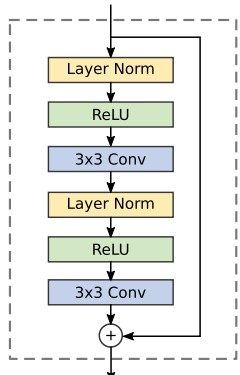

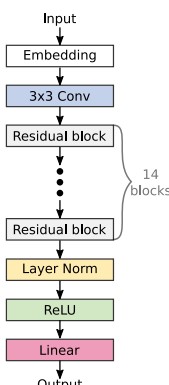

Figure 10: Residual block used in ResNet-29.

Figure 11: ResNet-29 full architecture for algorithmic tasks on matrices.

## D CONVERGENCE SPEED

Here we report Matrix Shuffle-Exchange (Matrix-SE) with 3.5M learnble parameters and ResNet-29 (ResNet) with 4.5M learnable parameters test error depending on the training step for algorithmic tasks. Both models were trained using Rectified Adam optimizer with learning rate 1e-4 and batch size 32. Fig. 12 and 13 report test error on matrix squaring (squaring), transpose (transpose), rotate by 90 (rotate90) degrees and bitwise XOR (bitwise XOR) tasks. Matrix Shuffle-Exchange converges significantly faster than ResNet-29 and achieves 0 error test rate on matrices of the same size as in training.

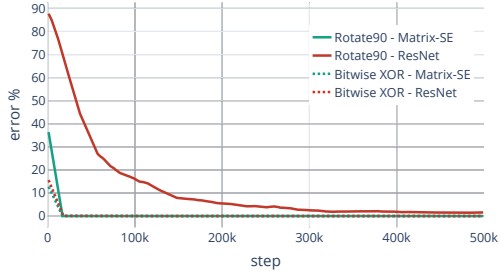
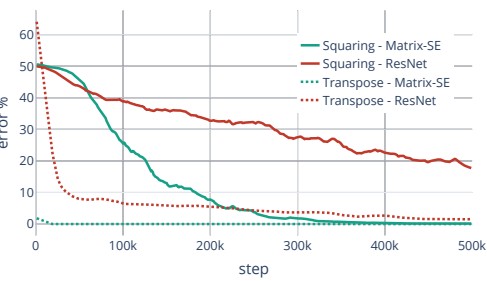

Figure 12: Error rate depending on training step on a test set for matrix rotation by 90 degrees and bitwise XOR operation.

Figure 13: Error rate depending on training step on a test set for matrix transpose and squaring.

## E   CIFAR-10 IMAGE CLASSIFICATION

We have evaluated Matrix Shuffle-Exchange performance on CIFAR-10 image classification task. CIFAR-10 (Krizhevsky et al., 2009) is a conventional image classification task that consists of 50000 train and 10000 test $32 \times 32$ images in 10 mutually exclusive classes. We use Matrix Shuffle-Exchange with 92 feature maps and 2 Beneš blocks. Between both blocks, dropout with rate 0.5 is applied. The model has a total of 3.5M learnable parameters.

We augment the training data by randomly flipping the input image and train Matrix Shuffle-Exchange for 200k iterations with batch size 32. The model without additional pretraining achieves 78% test accuracy, which is on par with a feed-forward network that is trained on a highly augmented dataset (Lin et al., 2015). But our model has somewhat worse accuracy than a typical fully-convolutional neural network, consisting of 9 convolutional layers that can achieve approximately 90% accuracy without additional pretraining (Hendrycks & Gimpel, 2016). The accuracy of our model could plausibly be improved using more augmentation or a larger dataset; otherwise, the model overfits.

