# OpenReview forum: "Matrix Shuffle-Exchange Networks for Hard 2D Tasks"
_ICLR.cc/2021/Conference — Reject_

### Official Review · AnonReviewer1 · 2020-10-28
**Simple model that performs well on logical reasoning tasks on 2D data, and generalizes to larger sizes than trained.**

**Rating:** 8
**Confidence:** 3

**Review:**

Summary: The paper proposes a network architecture called Matrix Shuffle-Exchange (Matrix-SE) that can learn many logical reasoning tasks on 2D data and graph. It has complexity O(n^2 log n) for 2D input of size n x n, which is much smaller than the complexity of naive attention applied to 2D data (O(n^4)). The proposed architecture is an adaptation of the Neural Shuffle-Exchange network architecture (Freivalds et al., 2019), moving from 1D to 2D data. This adaptation is done by using a Z-order iteration of the 2D input, then performing radix-4 shuffle and radix-4 exchange, instead of radix-2. This model is shown to be able to solve several hard tasks on 2D data, such as inferring algorithms on binary matrices (transpose, rotation, bitwise XOR, matrix squaring), graph operations (component labeling, triangle finding, transitivity), and solving Sudoku puzzles. The experiments show impressive results and the model's ability to generalize to test inputs of larger sizes than those in the training set.

======

Strengths:
1. Wide variety of algorithmic and logical reasoning tasks. I enjoyed reading the experiment section, and the tasks are fun and creative.
2. Impressive generalization on larger sizes. On those tasks mentioned above, the Matrix-SE model generalizes well to 2D arrays that have larger sizes than those in the training set, out-performing baselines (ResNet, Graph Isomorphism Network).
3. Simple model design. The generalization from Neural Shuffle-Exchange network to Matrix-SE is natural and straightforward.
4. The paper is well-written and easy to read.

Weaknesses:
1. Lack of theoretical characterization of the model. More concretely, what kind of operations can be represented by Matrix-SE? The theoretical motivation seems to be from the classic result that Benes networks can represent any permutation. However, it's not clear how expressive the proposed model is.
2. More realistic tasks. Do the logical reasoning tasks in more realistic scenario, such as modeling social networks represented as graphs?

======

Overall, I vote for accepting. The proposed model is simple, can perform logical reasoning tasks on 2D data, and generalizes well beyond sizes that are in the training set.

======

Additional feedback and questions:
- Do all the matrices in section 5.1 have binary values?
- How is accuracy defined in Table 1 and 2. Does the output matrix have to match the label exactly? Or is it accuracy per element?
- In Section 5.4, why is Residual-SE so much slower (9x) than Matrix-SE? I think it should only be 2x slower, because the depth of Residual-SE is twice that of the Matrix-SE.

==== After rebuttal: Thank you for clarifying details. I maintain my rating.
Showing that the inductive bias from Z-order curve and shuffle-exchange allows generalization to larger sizes is very interesting. Overall I vote for accepting.

---

> ### Author Response · Authors · 2020-11-12
> **Answer to questions**
>
> For algorithmic tasks (section 5.1), we use binary matrices only for XOR and matrix squaring tasks. For the rest of the tasks (transpose and rotation), we use integer values in range 1-11 (0 is reserved for padding).
>
> For both - matrix and graph - algorithmic tasks (Table 1. and 2.) we report accuracy per element, we have added explanations to the table description in the paper.
>
> You are right that the depth of Residual-SE is twice that of the Matrix-SE and theoretically, it should be only 2x slower, yet in our measurements (using the same Tensorflow version, computer and GPU) it performed 9x slower. That could be the result of various internal Tensorflow or CUDA optimizations, that favours shallower models.

---

### Official Review · AnonReviewer3 · 2020-10-28
**Interesting direction, but more work needed**

**Rating:** 4
**Confidence:** 3

**Review:**

This paper adapts the recently introduced Neural Shuffle-Exchange (NSE) network for 2D inputs. This is done by introducing two changes; flattening 2D input using a z-order curve, and using 4-to-4 switches rather than 2-to-2. Experiments on synthetic data show that the proposed model can outperform baselines.

### Strengths

1. The extension of NSE to 2D domains is an interesting research direction.
2. The use of the Z-order curve to map from 2D to 1D such that locality is approximately preserved is interesting.
3. The model performs well for the tasks explored here.

### Weaknesses

1. The overall contributions are limited. While flattening 2D data using a z-order curve is interesting, it is a well known technique for spatial indexing. The use of z-order curves is also not entirely unprecedented in the field. For example, [1, 2] also use z-order curves in the context of 2D data, and [3] in the context of 3D data.
2. The paper states that naive flattening of 2D data is expensive since it results in long sequences that are very slow to process with a NSE. The paper then proposes a technique which flattens 2D into an equally long sequence. This is very confusing. I'm assuming that reported speed improvements are due to the use of 4-to-4 rather than 2-to-2 switch, is this correct? Wouldn't this technique also be applicable to the original NSE?
3. The experiments are limited, consisting exclusively of synthetic data and a single baseline for each task. While an experiment involving images (CIFAR-10) is presented in the Appendix, the description is brief, and the model performs no better than a feed-forward network. This is concerning as CV seems like a natural domain for exploration, especially given overlapping motivation  between this work and recent literature exploring transformers in the vision domain [4, 5]. As such, while the results are promising, they are not enough to convince me of that Matrix-SE is likely to be useful for practical  problems of interest.
4. The presentation of technical and experimental details could be improved.
   1. I'm confused about how weights are shared. In particular, while the figure make it clear that weights are shared among Shuffle layers in the same block (similarly for Inverse Shuffle layers), I'm unsure if they are shared between units, although I suspect they are not.
   2. The original NSE paper pads "the input sequence to that length by placing the sequence at a random position and adding zeros on both ends." Is a similar technique used in this work? If not, and weights are not shared between units (Point 4.1), I'm confused about how the model can generalize to larger matrices.
   3. The paper does not contain information about how hyper-parameters were tuned, or information about stability of results. How robust is the model to different hyper-parameter settings? Are reported metrics averaged over multiple runs?
   4. It should be clarified in the paper that Z-Order curves are only approximately locality preserving and contain large discontinuous jumps.
   5. In section 5.2, the paper states that "Graph Isomorphism Network struggles to solve these tasks." However, on the hardest task, GIN outperforms Matrix-SE on triangle-finding on the largest graphs.

### Recommendation

I recommend rejection. While I believe the extension of NSE to 2D domains is a worthwhile pursuit, I do not believe this paper represents significant progress is this regard.

### Minor Issues

1. Section 5.1: "All tasks, except matrix squaring, **has** simple O(n2) algorithms." => "All tasks, except matrix squaring, **have** simple O(n2) algorithms."
2. Section 5.2: "a common **practise**" =>  "a common **practice**"
3. Inconsistent spacing is used around parenthesis.

### References

1. Zhang, Jianjin, et al. "Z-order recurrent neural networks for video prediction." 2019 IEEE International Conference on Multimedia and Expo (ICME). IEEE, 2019.
2. Kumar Jayaraman, Pradeep, et al. "Quadtree convolutional neural networks." *Proceedings of the European Conference on Computer Vision (ECCV)*. 2018.
3. Corcoran, Thomas, et al. "A spatial mapping algorithm with applications in deep learning-based structure classification." arXiv preprint arXiv:1802.02532 (2018).
4. Parmar, Niki, et al. "Image transformer." *arXiv preprint arXiv:1802.05751* (2018).
5. Child, Rewon, et al. "Generating long sequences with sparse transformers." *arXiv preprint arXiv:1904.10509* (2019).

---

> ### Author Response · Authors · 2020-11-12
> **Response to comments**
>
> We also initially thought CV as a good application domain but after extensive experiments, we found that convolutional networks have better inductive biases for images and long-range dependencies are structured and easily handled by the existing multiresolution approaches. Our model deals easily with image data but is behind the elaborated state-of-the-art models for CV tasks. We see applications of our network with other kinds of 2D data such as numerical spreadsheets, matrices of sensory measurements, dense graphs, etc. We did our best to find datasets for such tasks but were able to come up only with those presented in the paper. The lack of real-life could be because DL and open data are not well established in these domains or because there was not a realistically applicable method to deal with such data. Now, with the proposed architecture, there is a good chance that such applications will arise. Our further work is in part devoted to showing useful applications of the Matrix-SE network in the bioinformatics area and designing better ways to solve the vehicle routing problem.
>
> To achieve the best possible results for all tasks, hyper-parameters for each model were tuned by hand using manual grid-search. Reported metrics on algorithmic tasks are an average of 5 consecutive runs. We have added this to the description of Table 1. and 2.
>
> Due to an error in the data generation function for the GIN model (dataset for Matrix-SE was not affected), GIN on Triangle Finding tasks indeed seemed to perform better than Matrix-SE. We have made corrections in Table 2. for GIN model and fixed the bug in code (see supplementary code graph_networks/datasets.py). Now we can see that Matrix-SE outperforms GIN also on the triangle finding a task.
>
> You have understood correctly that speed improvements come from 4-to-4 Switch unit and radix-4 Shuffling layers, that results in $2\times$ fewer layers compared with standard NSE. One could speed up NSE in such a fashion, paying for it with the need to pad the input sequence to the closest length of $4^k$ instead of $2^k$.
>
> As you are correct that the weights of Switch Layers (and similarly Inverse Shuffle Layers) are shared for the same block. To achieve generalization on larger input instances we also share weights between the Switch Units of the same Switch layer. We have updated paper to emphasize the weight sharing between the units in the single Shuffle layer.

---

> > ### Comment · AnonReviewer3 · 2020-11-24
> > **Reply to Authors**
> >
> > Thank you for taking the time to address my questions and revise the paper. While some issues have been addressed (weight sharing, source of speedup), I am unable to raise my score at this time. I list my two primary concerns below.
> >
> > 1.  Using the z-order curve to flatten input is a fairly incremental contribution and the motivation is unclear. For example, in the author's common response they state "the purpose of the Z-Order curve is not to preserve locality." However, Section 4 of the paper states "raster scan flattening doesn’t preserve element locality if the inputs can be of different sizes."
> > 2. Given the incremental nature of the proposed modification, experiments on synthetic data alone are not enough. The author's state in their response that 2D datasets are not common. I do not agree with this statement. For example, [1], [2] present benchmark datasets for graph neural networks which include real-world data. Images are another obvious example, and while I understand the proposed architecture would likely be outperformed by SOTA ConvNets, I would expect an architecture specifically designed for 2D data to outperform a simple feedforward neural networks. Lastly, the authors state their architecture may be appropriate for "numerical spreadsheets." I find this confusing as a typical spreadsheet does not have meaningful 2D structure, i.e., the ordering of the rows and columns is arbitrary.
> >
> > ### References
> > 1. Dwivedi, Vijay Prakash, et al "Benchmarking graph neural networks." arXiv preprint arXiv:2003.00982, 2020.
> > 2. Errica, Federico, et al. "A Fair Comparison of Graph Neural Networks for Graph Classification." ICLR, 2019.

---

### Official Review · AnonReviewer5 · 2020-11-08
**Official Blind Review #5**

**Rating:** 4
**Confidence:** 3

**Review:**

This work proposes the Neural Shuffle-Exchange Network to capture both local and global dependencies for 2D data. The idea extends the 1D Neural Shuffle-Exchange Network to its 2D application. The proposed method first converts 2D data to 1D following the Z-order, then apply several Quaternary Switch and Quaternary Shuffle layers, and finally convert the data back to 2D space. The experimental results show that the proposed method can obtain better performance with reasonable computational cost.

Strengths:
+ The proposed method is very interesting. It studies how to capture long-term dependencies for 2D data in an efficient way.
+ It uses the Z-order curve to flatten 2D data into 1D. It can preserve locality information for features.
+ The experimental results show the proposed method can obtain better performance.

Weaknesses:
-  This work extends the Neural Shuffle-Exchange Network 1D to 2D. Compared with the 1D NSE, its technical contribution is incremental, and the novelty is limited.
-  Even though the Z-order curve can locality information, it is not clear whether the spatial information in 2D data can be preserved. It should be discussed.
-  The attention mechanism can learn non-local dependencies effectively. It should be compared in the experimental studies. In addition, it is true that the vanilla version of attention has O(n^4) complexity but there are several improved variants with competitive performance but lower computational cost.
- The proposed method is also applied to graph data and is compared with GIN. However, for graph data, I would suggest conducting experiments on benchmark datasets.

=====Update after rebuttal=====

I have read the authors' rebuttal. I still believe the novelty is limited, and hence I keep my score unchanged.

---

> ### Author Response · Authors · 2020-11-12
> **Comment about attention mechanisms**
>
> That's true that various sparse attention mechanisms exist and would be worth the comparison. Unfortunately, to the best of our effort, we were not able to find an attention or Transformer model that is suitable for our proposed 2D-to-2D tasks. We can see from recent efforts [1, 2, 3] that 1M input elements are still too many for most of the attention mechanisms (including the sparse ones).
>
> References:
> 1. Kitaev, N., Kaiser, Ł., & Levskaya, A. (2020). Reformer: The efficient transformer. arXiv preprint arXiv:2001.04451.
> 2. Parmar, N., Vaswani, A., Uszkoreit, J., Kaiser, Ł., Shazeer, N., Ku, A., & Tran, D. (2018). Image transformer. arXiv preprint arXiv:1802.05751.
> 3. Child, R., Gray, S., Radford, A., & Sutskever, I. (2019). Generating long sequences with sparse transformers. arXiv preprint arXiv:1904.10509.

---

### Author Response · Authors · 2020-11-12
**Common response**

We thank all reviewers for the valuable feedback!

Our observation is that most of the reviewers have expressed concerns about the Z-Order curve ability to preserve element locality.  In our case, the purpose of the Z-Order curve is not to preserve locality. It plays an integral role in our model, as it naturally divides elements of the input matrix into $2 \times 2$ element blocks and assigns each block to a single 4-to-4 Switch Unit. In this way, we achieve that our model can generalize to larger 2D input instances where the Z-Order curve serves as a natural inductive bias for this ability. Generalization capabilities are depicted in Fig.6, where we can see that the version with Z-Order curve generalizes, but the linear arrangement of values does not.

In this revision, we have done the following changes in paper and supplementary material:
* section 5.1: changed "All tasks, except matrix squaring, **has** simple O(n2) algorithms." to "All tasks, except matrix squaring, **have** simple O(n2) algorithms.";
* section 5.2: changed "a common **practise**" to "a common **practice**";
* fixed inconsistent spacing is used around parenthesis;
* fixed bug in data generator for Triangle Finding task for GIN model (supplementary code /graph_networks/dataset.py). Updated corresponding results in Table 2. GIN/ Triangle Finding;
* updated description of 1. and 2.Table with information that accuracy is per element and reported accuracy is average of 5 runs;
* section 4: added explanatory note about weight sharing between Quaternary Switch Units of the same Quaternary Switch layer.

We have addressed your questions and recommendations in subsections under your comments.

---

### Decision · Program_Chairs · 2021-01-07
**Final Decision**

**Decision:**

Reject

**Comment:**

The reviewers are split.  Two reviewers consider the technical contribution of the paper to be insufficient, and raise concerns about comparisons with Transformers or using more standard benchmarks for GNN experiments.   The other considers the experiments convincing and the method worth publishing.   My own view is that this work is not ready for inclusion in the conference.  In particular, I think this paper would be much stronger with either:

1: a more practical task to illustrate where this method might be applied in earnest,
2: more analysis and baselines on the synthetic data.  Synthetic data can be enough for a new method if it illuminates the functioning and the benefits and drawbacks.  In this paper, we have synthetic data with little analysis, and imo (concurring with R5) insufficient baselines.  For example, while a vanilla Transformer probably could not do the matrix problems (with the matrices encoded naively), one might expect Transformers with sparse attention to do quite well on e.g. transpose and 90 degree rotation, especially given the training curriculum and proper positional embeddings; a convolutional network seems like a strawman.  I also agree with R5 that standard benchmarks for GNN exist, and these might be appropriate (or at least there should be some discussion of why they are not).
 3: some theoretical discussion of what the proposed model can do that other methods fundamentally cannot.

I do think this is interesting work, and encourage the authors to revise and resubmit.